# Antimicrobial Activity of Quercetin, Naringenin and Catechin: Flavonoids Inhibit *Staphylococcus aureus*-Induced Hemolysis and Modify Membranes of Bacteria and Erythrocytes

**DOI:** 10.3390/molecules28031252

**Published:** 2023-01-27

**Authors:** Artem G. Veiko, Ewa Olchowik-Grabarek, Szymon Sekowski, Anna Roszkowska, Elena A. Lapshina, Izabela Dobrzynska, Maria Zamaraeva, Ilya B. Zavodnik

**Affiliations:** 1Department of Biochemistry, Yanka Kupala State University of Grodno, 230030 Grodno, Belarus; 2Laboratory of Molecular Biophysics, Department of Microbiology and Biotechnology, Faculty of Biology, University of Bialystok, 15-245 Białystok, Poland; 3Laboratory of Bioanalysis, Faculty of Chemistry, University of Bialystok, 15-245 Białystok, Poland

**Keywords:** flavonoids, antimicrobial agents, *Staphylococcus aureus*, hemolysis, membranes, fluidity

## Abstract

Search for novel antimicrobial agents, including plant-derived flavonoids, and evaluation of the mechanisms of their antibacterial activities are pivotal objectives. The goal of this study was to compare the antihemolytic activity of flavonoids, quercetin, naringenin and catechin against sheep erythrocyte lysis induced by α-hemolysin (αHL) produced by the *Staphylococcus aureus* strain NCTC 5655. We also sought to investigate the membrane-modifying action of the flavonoids. Lipophilic quercetin, but not naringenin or catechin, effectively inhibited the hemolytic activity of αHL at concentrations (IC_50_ = 65 ± 5 µM) below minimal inhibitory concentration values for *S. aureus* growth. Quercetin increased the registered bacterial cell diameter, enhanced the fluidity of the inner and surface regions of bacterial cell membranes and raised the rigidity of the hydrophobic region and the fluidity of the surface region of erythrocyte membranes. Our findings provide evidence that the antibacterial activities of the flavonoids resulted from a disorder in the structural organization of bacterial cell membranes, and the antihemolytic effect of quercetin was related to the effect of the flavonoid on the organization of the erythrocyte membrane, which, in turn, increases the resistance of the target cells (erythrocytes) to αHL and inhibits αHL-induced osmotic hemolysis due to prevention of toxin incorporation into the target membrane. We confirmed that cell membrane disorder could be one of the direct modes of antibacterial action of the flavonoids.

## 1. Introduction

Increased incidence of infections and considerable growth of resistance of microorganisms to antibiotics, along with production of resistant bacterial strains that substantially complicate treatment, are recognized as a very important global socio-medical problem. Thus, it seems to be of particular relevance to seek compounds that can be used either as an alternative or supplement to antibiotics [1]. The range of such antimicrobial agents also comprises such natural substances as flavonoids [2,3,4]. Bioactive polyphenols, flavonoids being among them, are essential secondary plant metabolites that are widely present in the human diet. By reducing oxidative stress, modulating cell signaling responses, regulating expression of certain genes or interacting directly with proteins and membranes, flavonoids play multiple important roles in plant defense and prevent numerous human pathologies with different mechanisms such as neurological diseases, cardiovascular diseases, infectious diseases and diabetes [5,6,7,8,9,10]. The potential of polyphenols (or their synthetic derivatives) as effective antibacterial and antiviral agents with a targeted mode of action, low side effects and the ability to avert bacteria toxin production and biofilm formation have been widely studied and well-documented [11,12]. 

Specific and nonspecific interactions of flavonoids with prokaryotic proteins and cell wall and membrane components influence bacterial growth, metabolism and pathogenesis, and it also plays an important role in the antibacterial effects of flavonoids [13]. These interactions include covalent bond formation, hydrogen bonding and hydrophobic interactions with bacterial cells. As was shown by proteomic analysis, the action of flavonoids perturbed different groups of bacterial enzymes involved in DNA-related metabolic processes (i.e., bacterial topoisomerases, helicases and DNA gyrases) and influenced ribosomal and membrane proteins, enzymes of cellular transport, bioenergy and lipid metabolisms, cell wall enzymes (i.e., efflux pumps and transporters, ATP synthase, cytochrome c and β-ketoacyl-acyl carrier protein synthases) and other targets [14,15]. These interactions impaired bacterial homeostasis, inhibited toxin and biofilm production and inhibited bacterial motility [14,15]. Recently Yaun et al. showed that the antibacterial activity of plant flavonoids against Gram-positive bacteria correlated with lipophilicity and that the bacteria cell membrane was the main site of flavonoid action [16]. The mechanism of the antimicrobial effect involved damage to phospholipid bilayers and inhibition of the bacterial respiratory chain or ATP synthesis [16].

The flavonol quercetin, one of the most abundant flavonoids, effectively inhibits growth of different drug-resistant Gram-positive and Gram-negative bacteria as well as fungi and viruses due to disruption of bacterial cell walls and membrane damage, intercalation of quercetin with DNA and inhibition of nucleic acid and protein synthesis, reduction of expression of virulence factors, inhibition of the activities of essential virulent enzymes and prevention of biofilm formation. Furthermore, specific structural modifications of quercetin have been shown to enhance its antimicrobial activity [12]. Unique antibacterial properties of quercetin in combination with antibiotics (oxacillin, ampicillin, vancomycin, gentamicin and erythromycin) against methicillin-resistant *S. aureus* were unraveled earlier. Simultaneously, quercetin induced morphological changes and aggregation of *S. aureus* cells, as assessed by electron microscopy [17,18,19]. In comparison, Wang at al. found that quercetin did not affect *S. aureus* strain (*S. aureus* Newman and its CoA mutant) viability but inhibited bacteria-secreted coagulase, which is essential for the virulence of *S. aureus*. Through this mechanism, quercetin inhibited *S. aureus* virulence. Furthermore, quercetin treatment reduced the retention of bacteria on catheter surfaces, decreased the bacterial load in the kidneys and alleviated kidney abscesses in vivo [20]. 

It was suggested that the antibacterial activity of another flavonoid, the flavanone naringenin, was dependent on the inhibition of the penicillin-binding protein PBP-2a [21]. The flavonol galangin, when combined with ceftazidime and amoxicillin, caused morphological damage to ceftazidime-resistant *S. aureus* and amoxicillin-resistant *E. coli* [19]. Most likely, the synergic effects of the above compounds were largely due to the variety of mechanisms of flavonoids and antibiotic effects on bacterial cells. In many cases, chalcones (bioprecursors of plant flavonoids) possessed stronger antibacterial activities in comparison with other classes of flavonoids. For example, 3-hydroxychalcone exhibited an approximately six-fold more pronounced effect on *H. influenza* than did the antibiotic azithromycin [22].

The antibacterial activity of flavonoids can be realized directly upon exposure of bacterial cells to the polyphenols, and it can be realized indirectly through effects on virulence factors of microorganisms. During the direct action of polyphenols on microorganisms, the mechanisms of antibacterial activity of flavonoids include [10,23,24,25,26,27,28]:Influences on the genetic apparatus, repression of genes or inhibition of nucleic acid synthases;Impairments of the membrane bilayer and membrane proteins, changes in membrane structure, fluidity and permeability, membrane pore formation and depolarization and ion leakage;Changes in bacterial metabolism, perturbation in bacterial homeostasis or enzyme inhibition (e.g., DNA gyrase inhibition);Inhibition of adhesions and microbial growth;Reactive oxygen species (ROS) generation;Controlling multidrug resistance, inactivation of bacterial efflux pump transporters (multidrug-resistance pumps) in bacteria and increasing susceptibility to antibiotics;Inhibition of binding to target cells;Metal ion chelation;Perturbation in cell envelope metabolism and envelope synthesis through the inhibition of fatty acids synthesis and;Damage to the bacterial respiratory chain, energy transduction mechanism uncoupling, inhibition of ATP synthase and disruption of bioenergetic status.

Recently, in addition to antibiotic approaches for combating bacteria, topical approaches have become a strategy for affecting virulent factors of microorganisms. This is based on interference with aspects of infectious pathogenesis by its efficient neutralization. It is suggested that such an approach will minimize the risk for drug resistance [29]. 

Flavonoids can also influence virulence factors, providing an antibacterial effect by:Prevention of toxin secretion, inactivation of toxins and bacterial lipopolysaccharides by interacting with the toxins and changing their conformation and activities [30];Inhibition of biofilm formation and cell adhesion by action on target cells to enhance their toxin resistance [31] and;Perturbation in organization of bacterial quorum and intercellular communication [32].

Recently, we analyzed the antihemolytic activity of four hydrolysable tannins, having different molecular masses and flexibilities, against two *Staphylococcus aureus* strains (8325-4 and NCTC 5655), and we showed a membrane-modifying action of the tannins. We found a correlation between the antihemolytic activity of the tannins studied and their capacity to increase the ordering parameter of the erythrocyte membrane and to change cell zeta-potential [29]. The antihemolytic effect of tannins from sumac leaves (*Rhus typhina* L.) under exposure to *S. aureus* seems to be largely associated with their effects on erythrocyte membrane structure. The sumac tannins incorporated into the erythrocyte membrane and induced transformation of discocytes into echinocytes, which enhanced the rigidity of the erythrocyte lipid bilayer and prevented membrane interaction with bacterial toxins [31].

Despite numerous investigations, the mechanisms of antibiotic and antivirulence activities of flavonoids require further research. For evaluation of antibacterial/antivirulence potential of some flavonoids, we used the clinically important *Staphylococcus aureus* strain NCTC 5655. *S. aureus* is a highly pathogenic Gram-positive, aerobic and toxin-producing microorganism, and it is one of the main microbial food contaminants [33] that causes a wide range of human and animal diseases [34,35]. The pathogenicity of *S. aureus* is largely due to its ability to secrete enzymes and toxins, primarily α-hemolysin (αHL) [29]. This αHL-secreting bacterial strain is widely used to understand the mechanism(s) of pathogenicity of bacteria and evaluation of the targets of antimicrobial drugs [2,3,15,17,20,29,34,36]. At the same time, due to vast genetic plasticity and strain variability, the specificity of quercetin and other flavonoids should be verified using other pathogens. αHL is a small, β-barrel, self-assembling, pore-forming toxin that has long been recognized as an important cause of tissue and cell damage during infection [37]. Bacterial cells secrete αHL, a mature, extracellular, water-soluble, monomeric protein of 293 amino acids (33 kDa). Seven αHL monomers are oligomerized under contact with host cell membranes. The αHL oligomer creates a hemolytic pore in the host cell membrane, which results in a loss of host cell integrity and cell death [38,39]. Recently, it was shown that some flavonoids (e.g., baicalin, kaempferol and quercetin) effectively inhibited the hemolytic activity of *S. aureus* αHL by modulating the expression of virulence factors [40]. In our experiments, we have shown for the first time another mechanism of quercetin’s antihemolytic activity, namely the ability to increase the resilience of the erythrocyte membrane to the toxin. The goal of this study was to compare the antihemolytic activity of flavonoids against hemolysis induced by α-hemolysin and the membrane-modifying action of flavonoids.

For analysis of structure–antibacterial activity relationships, we compared three molecules of flavonoids possessing high biochemical activity and representing the different most abundant classes: quercetin (flavonols), catechin (flavanols or flavan-3-ols), and naringenin (flavanones). To characterize the influence of the flavonoids on biophysical properties of erythrocytes and *S. aureus* bacteria, we measured cell diameter, zeta-potential, membrane organization and fluidity using fluorescence probes as well as electrokinetic and lipid bilayer techniques. To determine the susceptibility of the bacterial strain to the applied flavonoids, the lowest concentrations of the flavonoids that prevented bacteria growth (MIC) were assayed. A hemolysis assay was used for evaluation of protective effects of the selected flavonoids under erythrocyte exposure to *S. aureus*. 

## 2. Results

The present work investigated the mechanism(s) of the antibacterial and protective effects against *S. aureus* of the three flavonoids, quercetin, naringenin and catechin (Figure 1), which differ in structure, the number of OH-groups and lipophilicity (lipophilicity refers to the tendency of a compound to partition between a lipophilic organic phase and a polar aqueous phase) (Table 1).

### 2.1. Antimicrobial Effects of the Flavonoids

The antimicrobial activity of the flavonoids studied against the *S. aureus* NCTC 5655 strain was evaluated as the minimum inhibitory concentration (MIC) that stops bacterial growth. As Table 1 shows, quercetin had the greatest inhibitory effect, whereas naringenin and catechin demonstrated somewhat weaker effects. Simultaneously, Table 1 presents some flavonoid molecular parameters (e.g., dipole moment and torsion angle) calculated using our quantum-chemical considerations.

### 2.2. Effects of Flavonoids on Diameter, Zeta-potential and Membrane Structure of the Bacterial Cell

It is known that changes in membrane structure and integrity play a crucial role in bacterial growth and survival [44,45]. To evaluate the interaction of the flavonoids with cell membranes of viable *S. aureus* cells, we carried out a comparative analysis of the effects of the flavonoids on the diameter, zeta-potential (Figure 2) and membrane organization (Figure 3) of bacteria.

The measurements of the diameters of *S. aureus* cells in the presence of naringenin, quercetin and catechin showed that the interaction of quercetin at a concentration of 100 µM (close to the MIC) with bacteria in the medium resulted in an increase in particle diameter (by 60%). The diameter changes in the presence of catechin and naringenin at concentrations of 10 to 100 µM were not statistically significant (Figure 2A). (We represented the changes in particle diameter in percent due to the high dispersity of the values.) We did not observe any statistically significant changes in bacterial zeta-potential in the presence of the flavonoids studied (Figure 2B).

Figure 3A,B show changes in fluorescence anisotropy (Rs/R_0_) of DPH and TMA-DPH probes incorporated into bacterial cell membranes in the presence of the flavonoids studied. This parameter characterizes the state of membranes in the inner hydrophobic and in the aqueous/membrane interface regions, respectively. In our experiment, lipophilic quercetin decreased the Rs/R_0_ parameter for both probes, which indicates increased fluidity of bacterial cell membranes at different depths [46,47]. Naringenin also, but to a somewhat lesser extent, reduced the value for DPH fluorescence anisotropy and did not affect the TMA-DPH fluorescence anisotropy value. We did not observe any effects of hydrophilic catechin on the fluorescence anisotropy of either probe.

### 2.3. Antihemolytic Activities of Flavonoids During Sheep Erythrocyte Hemolysis Caused by S. aureus (NCTC 5655 Strain) and Flavonoid Effects on Erythrocyte and Liposomal Membrane Structures

Antihemolytic activities of the flavonoids were evaluated using the *S. aureus* NCTC 5655 strain, which secretes only αHL toxin [48], and sheep erythrocytes, which manifest high sensitivity to αHL [49]. Preliminary experiments showed that the flavonoids did not induce erythrocyte hemolysis in the range of the concentrations used (1 to 75 µM). The exposure of erythrocytes to NCTC 5655 cells caused effective hemolysis. As Figure 4 shows, quercetin, but not naringenin or catechin, effectively and dose-dependently prevented *S. aureus*-induced hemolysis.

Since the experimental conditions included the presence of quercetin in the medium during hemolysis, the antihemolytic activity of the polyphenol could be associated with neutralization of the αHL toxin itself and/or with modification of the erythrocyte membrane structure, which can limit the effect of αHL on erythrocytes. It is known that αHL oligomerization in the host cell membrane results in pore (transmembrane channels) formation and induced lysis of erythrocytes or other cells [50].

The antihemolytic activity of quercetin was studied using erythrocytes preincubated with polyphenol and subsequently washed with PBS (Figure 5).

The subsequent erythrocyte washing after preliminary exposure to the flavonoid did not remove the tightly bound lipophilic flavonoid and did not change the antihemolytic efficiency of quercetin (naringenin and catechin did not inhibit hemolysis). Quercetin molecules incorporated into host cell membranes, thus changing the membrane properties and preventing the hemolytic activity of αHL. 

To gain a deeper insight into the protective effect of the flavonoids on erythrocytes, we studied the effect of these compounds on the structural organization of the erythrocyte membrane by measuring the fluorescence anisotropy values for the DPH and TMA-DPH probes, which differ in their membrane localization (Figure 6).

Using the DPH probe, it was found that quercetin, naringenin and catechin, at low concentrations of up to 1 µM, enhanced fluorescence anisotropy of the probe located in the inner region of the erythrocyte membrane, which is indicative of increased rigidity of the membrane hydrophobic region (Figure 6A). The effect increased in the order of hydrophilic catechin < naringenin < lipophilic quercetin. Applying the TMA-DPH probe, we showed that quercetin, at a concentration of up to 1 µM, diminished probe fluorescence anisotropy, while naringenin and catechin did not significantly influence the membrane fluidity (Figure 6B). In the fluorescence experiments, we used the ratio of the [flavonoid]/[erythrocytes], similar to that in the hemolysis experiments.

The DPH probe, incorporated into the inner part of unilamellar liposomal DMPC vesicles, was also used for assessment of the penetration of the flavonoids into the lipid bilayer. Figure 7 represents the Stern–Volmer plots of the DPH fluorescence quenching by the flavonoids. The calculated Stern–Volmer constants, Ksv, which show the affinity of the flavonoids for the bilayer, are listed in Table 1. Ksv correlated with water solubility of the flavonoids (Table 1).

## 3. Discussion

The flavonoids naringenin, quercetin and catechin possess a wide range of biological and pharmacological activities, including their antioxidant, antibacterial, anti-inflammatory, cytoprotective, antitumor and antidiabetic actions [51,52]. At the first step of our work, we studied the effect of naringenin, catechin and quercetin on the NCTC 5655 *S. aureus* strain cell viability and showed that the flavonoids (MIC values were 100–200 µM, Table 1) exerted antibacterial effects. The antibacterial activities of numerous flavonoids were extensively evaluated previously. Table 2 lists the minimal inhibitory concentrations (MICs, the lowest concentrations of substances that prevent visible growth of bacterial strains) of the flavonoids. As can be seen, the MIC values obtained by the authors vary considerably. 

Numerous investigations have previously demonstrated that flavonoids effectively reduced bacteria adhesion, changed membrane stability and permeability, reversed antibiotic resistance and improved antibiotic efficacy. As was shown in the case of resistant strains of Staphylococcus spp., quercetin, at sub-inhibitory concentrations, prevented biofilm formation [54]. It was suggested that hydroxylation of the C5, C7, C3′ and C4′ positions of flavonoid molecules as well as geranylation or prenylation at C6 increased the antibacterial effect. On the other hand, the 5′-OH group (as in quercetin) or methoxylation at C3′ and C5 positions has been reported to decrease flavonoids’ antibacterial action [27,36,58]. 

A recent study showed high efficiency of myricetin as an αHL inhibitor [59]. The flavonoid simultaneously reduced the amount of *S. aureus*-produced αHL and neutralized its activity by interfering with αHL polymerization [59]. Lin et al. showed that flavonoids considerably suppressed growth of *S. aureus* or *E. coli* and noticed changes in activities of genes responsible for metabolic control and genetic information processing [24]. Decreased activities of ribosomal proteins [60], impaired synthesis of nucleic acids and inhibition of bacterial topoisomerases by polyphenols were found [61,62]. It was established that flavonoids were capable of enhancing the permeability of *S. aureus*, *E. coli* and *P. aeruginosa* membranes to ions [3,24]. Cushnie and Lamb described leakage of potassium in galangin-induced damage of the *S. aureus* cytoplasmic membrane [3]. In their earlier work, Ikigai et al. evaluated the antibacterial effects of green tea (*Camellia sinensis*) extracts, (–)-epigallocatechin gallate and (–)-epicatechin and demonstrated that the strong bactericidal action of (–)-epigallocatechin gallate caused considerable damage to phosphatidylcholine liposomes. A correlation between the antibacterial activity of catechins against *S. aureus* and *E. coli* and their damaging effect on liposomal membranes was shown [63]. Eumkeb and Chukrathok related apigenin- and naringenin-induced changes in the *Enterobacter cloacae* cytoplasmic membrane to inhibition of energy metabolism and metabolic disorders [19]. Haraguchi et al. showed that chalcones isolated from roots of *Glycyrrhiza inflata*, which demonstated antimicrobial activity, effectively inhibited NADH cytochrome c reductase and oxygen consumption in susceptible bacterial cells. The site for respiratory inhibition was thought to be between CoQ and cytochrome c in the bacterial respiratory chain [64]. Mirzoeva and coauthors found uncoupling of the energy-transducing cytoplasmic membrane and inhibition of the proton-driving force and cell motility of *S. aureus* by an ethanolic extract of propolis and its cinnamic and flavonoid components (quercetin). These effects on the bioenergetic status of the membrane may contribute to the antimicrobial action of propolis [65]. Li and coauthors found that procyanidins elevated the activities of Na^+^/K^+^-АTPase and Ca^2+^-ATPase and decreased those of malate dehydrogenase and adenosine triphosphatase in *S. aureus* [66]. 

The capacity for neutralizing bacterial toxins is an important property of flavonoids. Flavonoids can affect different virulence factors and are capable of neutralizing pore-forming toxins and blocking/modifying the channels formed by them in the membranes of host cells. The antivirulence activities of flavonoids with respect to staphylococcus enterotoxin A depended on the polymerization degree of the flavonoids [67]. High-molecular-mass polyphenols form aggregates with αHL, inhibiting its effect on erythrocytes [68]. An antihemolytic effect was observed when polyphenol-pretreated erythrocytes were washed before addition of *S. aureus*-produced αHL, which agrees with our results and confirms the capacity of plant polyphenols to produce antihemolytic activities at the level of erythrocyte membranes [68]. 

Polyphenols were demonstrated to inhibit the anion-selective and permeable to urea channel formed by vacuolating cytotoxin (VacA) produced by *Helicobacter pylori* [69] and modify the conductivity of the pore formed by *S. aureus* αHL [70]. It was established that proantocyanidines were capable of detoxifying lipopolysaccharide, the endotoxin of Gram-negative bacteria [71]. The polymeric fraction of black tea thearubigin (polymers of tea catechins) blocked the neuromuscular effects of neurotoxins A, B and E produced by Clostridium botulinum due to catechin chelation of the metalloproteinase moiety of the toxin [33,72]. Black tea caempherol and quercetin glycosides also inhibited the toxic effects of botulotoxins [33]. Morianga et al. reported that polyphenol compounds inactivated the virulence of cholera toxin [73]. The antihemolytic activity of polyphenols may be associated with their effect on bacterial metabolism, which causes a diminution of the amount of toxins released, as was shown earlier for (–)-epigallocatechin [74].

At the next step, we analyzed the influence of the flavonoids on the bacterial cell membrane properties as one of the possible mechanisms of the antimicrobial effects of the flavonoids against αHL-producing NCTC 5655 *S. aureus*. Quercetin (100 µM), but not naringenin or catechin, increased the bacterial cell nanosize measured, probably as a result of cell aggregation (Figure 2A). In our previous experiment, the incorporation of the flavonoids quercetin and naringenin (but not catechin), into the liposomes enlarged the area of the bilayer [39]. He and coworkers have recently shown that quercetin, at sub-MIC concentrations, considerably increased the aggregation rate of *Porphyromas gingivalis* and cell surface hydrophobicity [57]. In our work, *S. aureus* zeta-potential, an electrochemical characteristic of the cell surface, was measured in PBS, pH 7.4, and was found to be –13.4 ± 1.7 mV. The effect of the flavonoids studied on bacterial zeta-potential, as a possible marker of changes in membrane permeability, was not statistically significant (Figure 2B).

In our work, quercetin (10–50 µM) significantly increased the fluidity of the outer layer and the surface area of bacterial cell membranes. In contrast, the antibacterial activity of 2R,3R-dihydromyricetin against *S. aureus* was related to flavonoid-induced morphological changes and a considerable reduction in fluidity of the membrane inner part (using the DPH probe), membrane hyperpolarization and disruption of membrane integrity [75]. Naringenin enhanced the fluidity of the surface area to a lesser extent, and hydrophilic catechin did not affect the fluidity of the bacterial membrane. Earlier we showed that quercetin (10–50 μM), rather than catechin and naringenin, strongly decreased the microfluidity of the liposomal membrane bilayer at different depths and hydration in the region of polar head groups [39]. The different effects of the flavonoids on fluidity of the surface area of cellular and liposomal membranes may be due to flavonoid interactions with the protein component of the biological (bacterial) membrane. A significant correlation between the antimicrobial activity of flavonoids and the liposomal membrane rigidification effect was found earlier, and it was suggested that the flavonoids exerted their antibacterial effect by reducing membrane fluidity [53]. 

In our experiments, by increasing the fluidity of the bacterial membranes (Figure 3), quercetin, but not naringenin or catechin, influenced *S. aureus* toxicity and inhibited the hemolysis of sheep erythrocytes induced by exposure to *S. aureus* (Figure 4). The antihemolytic effect did not attenuate after washing erythrocytes preliminarily exposed to the flavonoid, as a result of tight binding of quercetin to the erythrocyte membrane (Figure 5). It should be noted that quercetin influenced bacterial membrane organization and inhibited *S. aureus*-induced hemolysis at concentrations lower than the average value for MICs (Table 1). The calculated IC_50_ value, corresponding to the concentration of quercetin that inhibits hemolysis by 50%, was 65 ± 5 μM for the *S. aureus* NCTC 5655 strain.

We demonstrated that the flavonoids (in the order of catechin < naringenin < quercetin) enhanced the stiffness of the erythrocyte membrane hydrophobic region. Simultaneously, quercetin, but not naringenin or catechin, increased the mobility of the erythrocyte membrane surface zone. We suggest that the antihemolytic effect of quercetin was associated with the effect of the flavonoid on the organization of the erythrocyte membrane lipid bilayer, which, in turn, seemed to prevent αHL-induced osmotic hemolysis. The fundamental step of pore formation consists of binding of toxin protomers to a receptor (lipids, glycans or proteins) on the surface of the target cell membrane. Receptors increase the local concentration of the toxin and also promote oligomerization [76]. As was also demonstrated previously, flavonoids could interact with phospholipid and protein components of erythrocyte membranes and protect erythrocytes against hypotonic lysis [77]. It was shown that the liquid disorder phase was preferable for binding of hemolysin to lipids and its oligomerization [78].

The mechanism of flavonoid interactions with membranes and their localization are fundamental to understanding of the pharmacological activities of flavonoids. Using the Stern–Volmer constant, Ksv, of DPH fluorescence quenching by the flavonoids in the liposomal membrane, we estimated the availability of the quencher (flavonoid) to the excited fluorophore (DPH). As Table 1 and Figure 7 demonstrate, quercetin effectively penetrates into the membrane interior. The accessibility of naringenin and catechin to the membrane interior was much lower, in accordance with flavonoid lipophilicity.

For the purposes of demonstration, we provided a molecular model of the structure of αHL consisting of seven polypeptide chains (X-ray structural analysis data were obtained from the Protein Data Bank [https://www.rcsb.org], accessed on 3 March 2021). The diameter of the pore inner cavity formed by αHL was ≈12 Å (Figure 8), whereas the linear sizes of the quercetin, catechin and naringenin molecules were equal to 11.81, 11.65 and 10.42 Å, respectively. (Figure 1). Despite similar linear sizes of the flavonoid molecules and pore diameter, only quercetin had an antihemolytic effect. Earlier, we demonstrated a planar structure of quercetin, but not of catechin or naringenin, and the optimal electron orbital delocalization due to the C2=C3 double bond in the C ring (absent in catechin and naringenin molecules) [79]. The planar geometry of the flavonoid molecule may be crucial for antibacterial activity [80]. Crystallographic data previously showed that *S. aureus* α-hemolysin, a cytolytic endotoxin, formed the mushroom-shaped homo-oligomeric heptamer, which was a solvent-filled transmembrane channel with a hydrophilic interior 100 Å in length and ranging from 14 Å to 46 Å in diameter [81]. Similarly, using penetrating water-soluble polymers, the authors estimated two practically identical openings of the channel with radii of 1.2–1.3 nm and two apparent constrictions with the radii of 0.9 nm and 0.6–0.7 nm, occurring in the channel lumen [82]. These findings correlated with the data indicating that the α-hemolysin channel had a wide vestibule leading into the pore with the diameter of 15 Å [83]. Different organic molecules could block a single αHL-formed pore [70,84]. The dipole moment and the net charge of penetrating molecules play an important role in the transportation through nanopores [83]. The dipole moment of lipophilic quercetin was the lowest, and the largest percentage of quercetin molecules were ionized at the pH of the medium in comparison with the other flavonoids studied. Ostroumova and coauthors showed that the flavonoids hydroxylated in position five of the A-ring and in position 4′ of the B-ring specifically interacted with the voltage sensor of the αHL pore and shifted the αHL-formed channel from a high- to low-conductance state [70]. Olchowik-Grabarek et al. demonstrated that iodogallic acid formed static complexes with α-hemolysin in solutions and inhibited its nanopore conduction in artificial lipid bilayers [85]. It was shown in the case of large tannins that more flexible molecules exerted a more pronounced antihemolytic effect [29].

Thus, summarizing the results obtained, we can conclude that the flavonoids effectively interacted with biological (bacteria and erythrocytes) and artificial membranes and changed the dynamic characteristics and organization of the membranes. Recently, it was demonstrated that subinhibitory concentrations of antibiotics may not alter membrane fluidity of bacterial cells [86]. In our experiments, quercetin (but not catechin or naringenin) exerted pronounced effects on the membrane structure, fluidity and size of bacterial cells. The high lipophilicity, the C2=C3 double bond and the planarity allowed the quercetin molecule to more effectively inhibit *S. aureus*-induced osmotic hemolysis. Recently, it was concluded that the C2=C3 double bond in flavonoid molecules enhanced Gram-positive bacterial inhibition [58]. It has already been shown that flavonoid molecules (such as quercetin), predominantly located in the hydrophobic region of the membrane bilayer, can initiate the formation of raft-like domains, while the flavonoid molecules located in the polar interface region of the bilayer can fluidize membranes [87]. A specific interaction between flavonoids and some integral membrane proteins is also possible [87]. The antihemolytic activity of quercetin may be due to stiffening of the inner hydrophobic part of the erythrocyte membrane and fluidization of the membrane surface, resulting in a decrease in host cell sensitivity to toxin-induced hemolysis. It is known that the membrane lipid phosphatidylcholine was originally proposed to be an α-hemolysin receptor due to the high-affinity binding observed between αHL and clustered phosphocholine head groups [88]. Previously, it was suggested that the interactions of tannins with the phosphate groups of phospholipids caused stiffening of the erythrocyte membrane outer layer, leading to restriction in the αHL incorporation into the membrane and limitation of the formation of a functional channel [29]. In our experiments, we used sheep erythrocytes that had not been found to possess any specific proteinacious receptors for αHL [29]. One of the possible antihemolytic mechanisms of flavonoids may be physical blocking of the αHL-formed pore. Molecular modeling showed a good agreement of the linear sizes of the polyphenols studied with the diameter of the αHL inner cavity. 

## 4. Conclusions

The emergence of microorganisms with multiple drug resistance and high virulence requires a search for new therapeutic strategies to treat resistant infections. To gain a deeper insight into their antibacterial and protective activities, we carried out a comparative analysis of the effects of flavonoids on the diameter, zeta-potential and membrane organization of viable *S. aureus* cells and on the structure of sheep erythrocyte membranes. The accessibility of naringenin and catechin to the membrane interior was much lower when compared with quercetin, in accordance with flavonoid lipophilicity, as revealed by fluorescence studies. In our experiments, naringenin, catechin and quercetin (100–200 µM) inhibited *S. aureus* cell growth depending on the distorting effect on the bacterial membrane structure. Only lipophilic quercetin at concentrations of 10–80 µM, and not catechin or naringenin, inhibited sheep erythrocyte hemolysis induced by the *S. aureus-*produced αHL toxin. The antihemolytic and membrane-modifying effects were observed at quercetin concentrations well below the minimal inhibitory concentration values for *S. aureus* growth. Using spectroscopic, electrokinetic and lipid bilayer techniques, we report a novel mechanism of antihemolytic activity of quercetin, which consists of enhancing resistance of target cells (erythrocytes) to αHL due to the membrane-modifying effect, restriction of interactions of erythrocytes with the toxin, and prevention of osmotic hemolysis. Previously, it was demonstrated that most antibiotics have intracellular bacterial targets [86]. We confirmed that alteration of bacterial and host cell membranes could be one of the direct modes of antibacterial action of flavonoids.

## 5. Materials and Methods

### 5.1. Materials

Naringenin (98%), product no. W530098; quercetin (≥95%), product no. Q4951; (+)-catechin hydrate (≥98%) product no. C1251; 1,6-diphenyl-1,3,5-hexatriene (DPH), 1-(4-trimethylammoniumphenyl)-6-phenyl-1,3,5-hexatriene (TMA-DPH) and 1,2-dimyristoyl-sn-glycero-3-phosphocholine (DMPC) were from Merck/Sigma-Aldrich (St Louis, MO, USA, or Steinheim, Germany). Mueller–Hinton (MH) agar and Mueller–Hinton (MH) broth suitable for microbiology were supplied by Oxoid (Basingstoke, UK). Other reagents and organic solvents were of analytical grade, purchased from POCh (Gliwice, Poland) and used without further purification. Sheep blood was obtained from GrasoBIOTECH (Graso Company, Starogard Gdański, Poland). The flavonoids were used as freshly prepared stock solutions (15 mM) in ethanol. In the preliminary experiments, we showed that ethanol, at the concentrations used (not exceeding 1.3%), did not considerably affect the parameters measured. 

### 5.2. Bacterial Strain and Growth Condition 

The *S. aureus* strain NCTC 5655, obtained from the National Collection of Type Cultures (Salisbury, England), was used. The bacteria were grown on Mueller Hinton (MH) nutrient agar plates. Before the experiments, the bacteria were grown overnight at 37 °C in Mueller–Hinton (MH) broth with 200 rpm shaking.

### 5.3. Antibacterial Activity of Flavonoids

The minimum inhibitory concentration (MIC), the lowest concentration of a substance that prevents the visible growth of bacteria, was determined by the following experimental procedure. *S. aureus* cells, strain NCTC 5655, were grown overnight at 37 °C in Mueller–Hinton (MH) broth with shaking at 200 rpm, and the cell suspension was adjusted to the absorbance OD_600_ = 0.1 at a wavelength λ = 600 nm. Then 180 μL of the obtained bacterial suspension was added to the wells of a sterile 96-well microtiter plate containing various concentrations of flavonoids in the range of 10–300 μM. Control wells did not contain flavonoids. The plate was shaken on a microplate shaker for 1 min and then was incubated for 24 h at 37 °C. The visible growth of bacteria was evaluated as an increase in OD_600_. The lowest concentration of flavonoids preventing bacteria growth was taken as its MIC. To evaluate the MIC, the samples were examined using a microplate reader SpectraMax M2 (Molecular Device, San Jose, CA, USA). To avoid the effect of flavonoids on turbidity/absorbance of the medium, negative controls were run for each flavonoid concentration. 

### 5.4. Inhibition of Sheep Erythrocyte Hemolysis by Flavonoids

A hemolysis assay was used for analysis of the protective effects of flavonoids. The hemolytic activity of *S. aureus* NCTC 5655 in the presence and in the absence of the flavonoids was measured using sheep erythrocytes as previously described by Olchowik-Grabarek et al. [30]. Erythrocytes were isolated from sheep blood by centrifugation (850× *g*, 15 min) and washed three times with isotonic buffered saline (PBS, 145 mM NaCl, 1.9 mM NaH_2_PO_4_, 8.1 mM Na_2_HPO_4_, pH 7.4) at 4 °C. Two milliliters of the erythrocyte suspension with 1% hematocrit were incubated at 37 °C for 30 min in the absence or presence of flavonoids (3.75–75 μM) in PBS buffer. In some experiments, erythrocytes were washed after preliminary exposure to the flavonoids. The *S. aureus* strain NCTC 5655 maintained in MH broth was adjusted to an optical density OD_600_ = 2.1 at a wavelength λ = 600 nm. Then 100 μL of bacteria in MH broth was added to each sample of erythrocytes. After incubation for 60 min at 37 °C, 0.5 mL of the suspension was taken from each sample and mixed with 1 mL of PBS buffer. To obtain 100% hemolysis, 0.5 mL of the suspension was mixed with 1 mL of distilled water. All the samples were centrifuged (850× *g*, 15 min, 4 °C, Hermle Z32 HK, Hermle Labortechnik GmbH, Wehingen, Germany), and the optical density of the supernatants was measured at 540 nm using a Jasco V-770 spectrophotometer (Jasco Corporation, Tokyo, Japan). The results are presented as a percent of hemolysis depending on the concentration of the flavonoids added.

### 5.5. Measurements of Erythrocyte Membrane Fluidity

Erythrocyte membrane organization was analyzed by fluorescence anisotropy of TMA-DPH and DPH probes (Merck/Sigma-Aldrich (St Louis, MO, USA, or Steinheim, Germany)), which differ in their membrane localization. A suspension of erythrocytes (3 mL of 0.01% hematocrit in PBS) was labeled with a fluorescent probe (DPH or TMA-DPH) at a concentration of 1 μM (20 min, 37 °C) in the dark. DPH was dissolved in tetrahydrofuran, and TMA-DPH was dissolved in methanol. The initial probe concentration was 1 mM. Fluorescence anisotropy values were recorded in the absence and presence of the flavonoids at 37 °C with a Perkin-Elmer LS 55B spectrofluorometer (Perkin-Elmer, Pontyclun, UK) equipped with a fluorescence polarization device. Changes in membrane fluidity after addition of the flavonoids (30 min, 37 °C) in the concentration range from 0.05 to 1.0 μM were determined based on the values for the fluorescence anisotropy of DPH or TMA-DPH probes (R), which were calculated by the fluorescence data manager program according to the standard anisotropy equation:

(1)$$r=\frac{I_{\mathrm{VV}}{\;–\;\mathrm{GI}}_{\mathrm{VH}}}{I_{\mathrm{VV}}{+2\mathrm{GI}}_{\mathrm{VH}}}$$where I_VV_ and I_VH_ are the vertical and horizontal fluorescence intensities, respectively. The factor G = I_HV_/I_HH_ (grating correction factor) corrects the polarizing effects of the monochromator. The excitation wavelengths were 348 nm for the DPH probe and 340 nm for the TMA-DPH probe, and the fluorescence emission was measured at 426 nm for the DPH probe and 430 nm for the TMA-DPH probe. The results are presented as a ratio (Rs/R_0_), where Rs is the fluorescence anisotropy of the probes in the presence of flavonoids and R_0_ is the fluorescence anisotropy of the probes in the absence of flavonoids. The DPH probe was localized in the hydrophobic region of the membrane occupied by carbohydrate chains of lipids, whereas the TMA-DPH chain was predominantly localized on the border of the external environment/membrane. The fluorescence anisotropy values of the probes characterized orderliness and motility of the lipid bilayer in the inner and surface membrane regions [29].

### 5.6. Measurements of S. aureus Membrane Fluidity

The structural organization of bacterial membranes was analyzed by the fluorescence anisotropy values of the DPH and TMA-DPH probes. The number of *S. aureus* cells was photometrically standardized. Cells (cell suspension OD_600_ = 0.01 in 10 mM PBS pH 7.4) were incubated with DPH or TMA-DPH at a final concentration of 1 µM for 15 min in the dark at 37 °C. After that, various concentrations of flavonoids (2.5–50 µM) were added to the bacterial cell suspension, and the cells were incubated for 45 min at 37 °C. The values for fluorescence anisotropy of the probes incorporated in the membranes were measured in the absence and in the presence of flavonoids using a Perkin-Elmer LS 55B spectrofluorometer (Perkin-Elmer, Pontyclun, UK). 

### 5.7. S. aureus Nanoscale Cell Diameter and Zeta-Potential

Zeta-potential, a tool for studying the alteration in bacterial cell surface permeability, and the average diameter of bacteria were measured using a Malvern Zetasizer Nano ZS (Malvern Instruments Ltd, Malvern, UK). Zeta-potential is an electrokinetic potential associated with the mobility of charged particles, and its changes make it possible to estimate electrostatic forces between interacting particles (bacterial cells). The *S. aureus* cell diameter (nanosize) was analyzed using dynamic light scattering (DLS). The bacterial cells (OD_600_ = 0.01) were suspended in PBS buffer, pH 7.4. The measurements were taken in disposable folded capillary cells (DTS 1070), and the data were analyzed using Malvern software (Malvern Instruments Ltd, Malvern, UK).

### 5.8. Interaction of Flavonoids with Liposomal Membranes

Using the Stern–Volmer constant, Ksv, of DPH fluorescence quenching by flavonoids in the liposomal membrane, we estimated the availability of the quencher (flavonoid) to the excited fluorophore (DPH) and, therefore, flavonoid penetration into the membrane interior. Liposomes (unilamellar bilayer vesicles) were prepared by an extrusion technique using an Avanti Polar Lipids Mini-Extruder (Birmingham, AL, USA) and were composed of 1,2-dimyristoyl-sn-glycero-3-phosphocholine (DMPC) (14:0), as described previously [89]. The liposomes (100 µg phospholipid/mL) were incubated with DPH at a final concentration of 1 µM for 20 min at 25 °C in PBS, pH 7.4, and fluorescence intensity values were registered in the absence and presence of flavonoids. The results are presented as a Stern–Volmer plot [(F_0_ − F)/F] = Ksv [Q], where Fo is the fluorescence intensity of the probe in the absence of flavonoids, and the F is the fluorescence intensity of the probe in the presence of flavonoids. Ksv is the Stern–Volmer constant, and [Q] is the flavonoid (quencher) concentration. The excitation and emission wavelengths were 348 and 426 nm, respectively.

### 5.9. Calculations of Flavonoid and S. aureus α-Hemolysin Molecular Geometries

Flavonoid molecules were theoretically considered by performing both the semi-empirical molecular orbital theory and ab initio calculations. The Austin Model 1 (AM1) semi-empirical method within unrestricted Hartree–Fock (UHF) formalism in the self-consistent field approximation and the Polak–Ribiere algorithm were considered to fully optimize the geometry of the flavonoid molecules [90]. We performed all the calculations by using the HyperChem-8.0 software package (HyperCube, Inc., Gainesville, FL, USA) for searching the conformations with minimum energy [http://www.hyper.com] (accessed on 17 March 2021). 

### 5.10. Statistics

All data are expressed as means ± SD for four to six experiments. Differences between parameter values measured in groups were analyzed using Student’s *t*-tests or non-parametric Kruskal–Wallis tests. The normality of distribution was determined by the Shapiro–Wilk test. Statistical analysis was carried out using the STATISTICA 6.0 software package (StatSoft, Inc. Tulsa, OK, USA). The level of significance was set at *p* < 0.05.

## Figures and Tables

**Figure 1 molecules-28-01252-f001:**
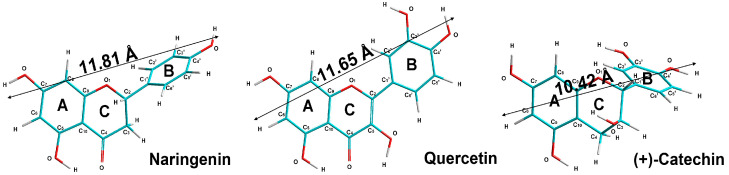
Optimized molecular structures of quercetin, catechin and naringenin and linear sizes of molecules (by ab initio method with 6-31G basis and unrestricted Hartree–Fock (UHF) method with Polak–Ribière gradient algorithm).

**Figure 2 molecules-28-01252-f002:**
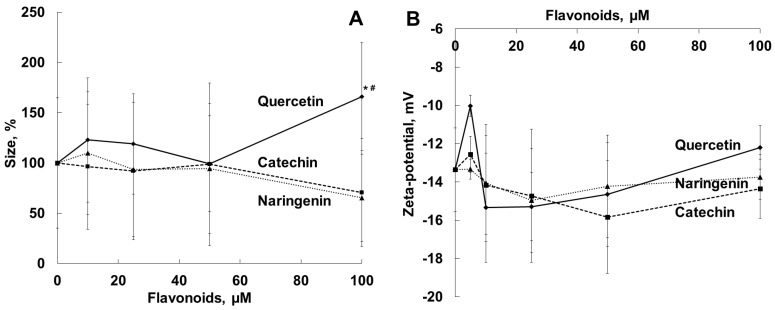
Effects of quercetin, catechin and naringenin on *S. aureus* cell diameter (**A**) and zeta-potential (**B**). Bacterial cells were suspended in PBS, pH 7.4. *—*p* < 0.05 in comparison with the cells in the absence of the flavonoids; #—*p* < 0.05 in comparison with the cells in the presence of other flavonoids.

**Figure 3 molecules-28-01252-f003:**
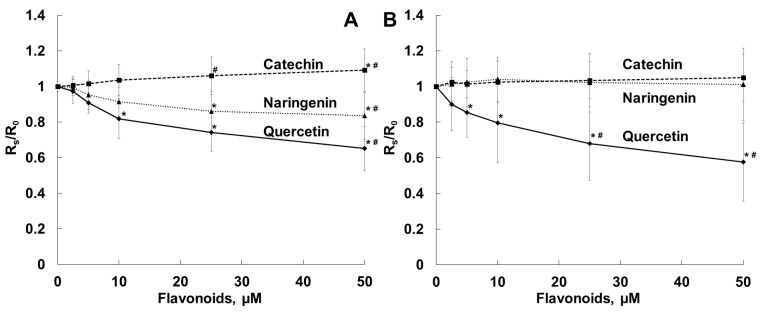
Dependence of fluorescence anisotropy of DPH (**A**) and TMA-DPH probes incorporated in bacterial membranes (**B**) on flavonoid concentrations. Bacterial cells were incubated with TMA-DPH or DPH at a final concentration of 1 µM in PBS, pH 7.4, and anisotropy values were recorded in the absence (R_0_) and in the presence (Rs) of the flavonoids. *—*p* < 0.05 in comparison with dye fluorescence anisotropy in the absence of the flavonoids; #—*p* < 0.05 in comparison with dye fluorescence anisotropy in the presence of other flavonoids.

**Figure 4 molecules-28-01252-f004:**
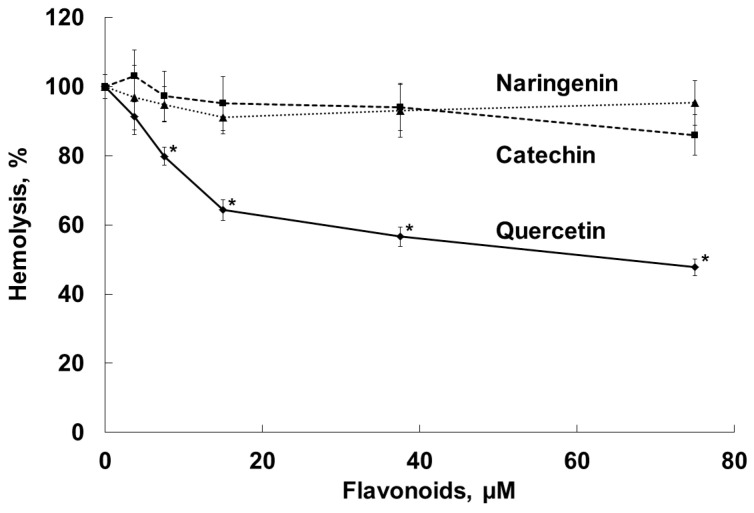
Inhibition by flavonoids (naringenin, quercetin and catechin) of sheep erythrocyte lysis induced by *S. aureus* NCTC 5655 in PBS, pH 7.4, 37 °C, 60 min. Erythrocytes (1% hematocrit) were preincubated with the flavonoids over 30 min, and then 100 µL of bacteria suspension in PBS buffer (OD_600_ = 2.1) was added. The results represent the average values ± standard deviation, n = 6. Hemolysis induced by *S. aureus* in the absence of flavonoids was taken as 100 %. *—*p* < 0.05 compared with erythrocytes in the absence of the flavonoids.

**Figure 5 molecules-28-01252-f005:**
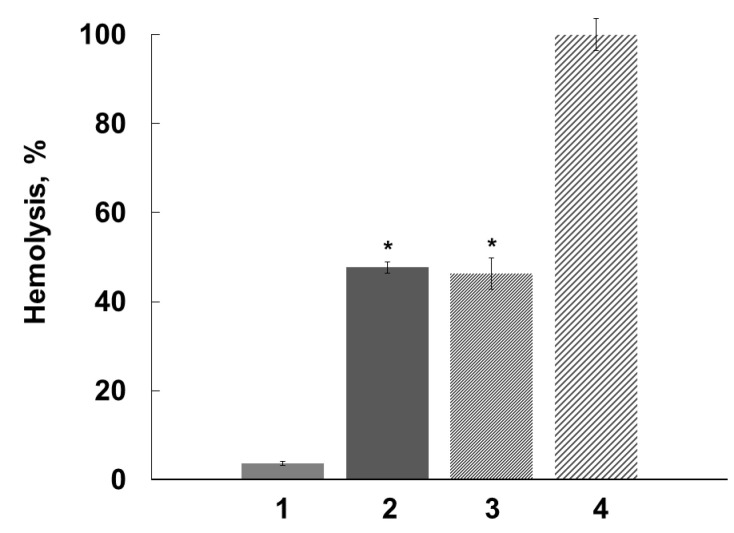
Inhibition by quercetin (75 µM) of sheep erythrocyte lysis induced by *S. aureus* NCTC 5655. 1—hemolysis in the absence of bacterial cells; 2, 3 and 4—hemolysis in the presence of bacterial cells in the medium; 2—hemolysis in the presence of quercetin; 3—hemolysis of erythrocytes washed after preliminary incubation with quercetin; 4—hemolysis induced by *S. aureus* NCTC 5655 in the absence of quercetin. The experimental conditions were similar to those shown in Figure 4. *—*p* < 0.05 in comparison with erythrocytes exposed to bacteria in the absence of flavonoids.

**Figure 6 molecules-28-01252-f006:**
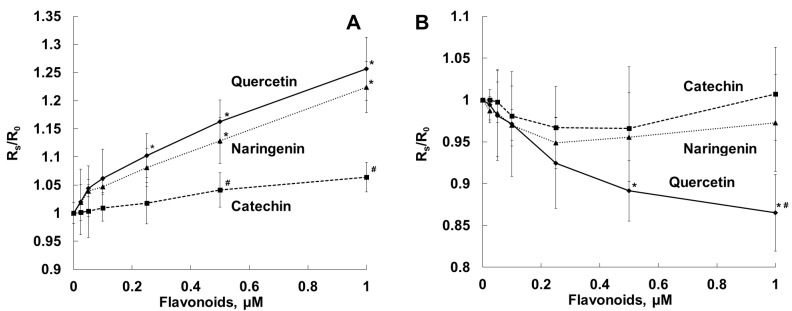
Dependence of fluorescence anisotropy of DPH, located in the hydrophobic region of the erythrocyte membrane (**A**), and TMA-DPH, located in the outer monolayer of the erythrocyte membrane (**B**), on the concentration of the flavonoids. Suspension of erythrocytes (0.01% hematocrit in PBS) was labeled with the fluorescent probes (DPH or TMA-DPH) at a concentration of 1 μM in the dark for 20 min at 37 °C, and then the flavonoids were added and incubated for 30 min at 37 °C. Anisotropy values were registered in the absence (R_0_) and in the presence (Rs) of the flavonoids. *—*p* < 0.05 in comparison with dye fluorescence anisotropy in the absence of the flavonoids; #—*p* < 0.05 in comparison with dye fluorescence anisotropy in the presence of other flavonoids.

**Figure 7 molecules-28-01252-f007:**
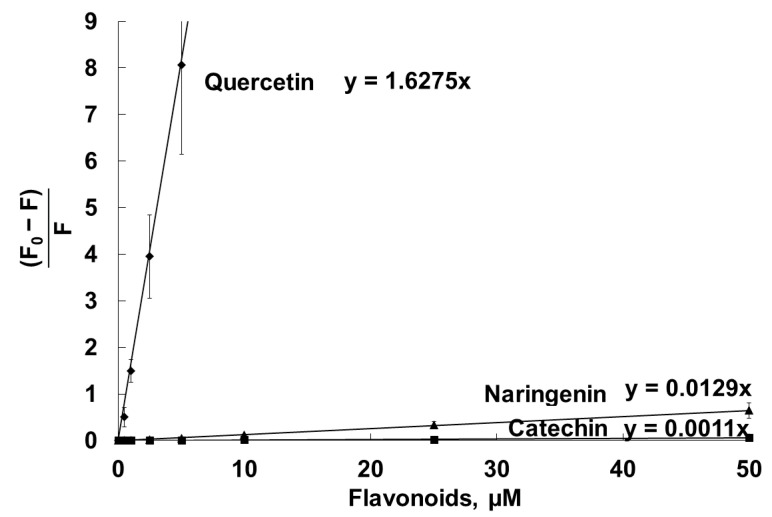
Stern–Volmer plots of quenching by catechin, naringenin and quercetin of the fluorescence of the DPH probe, located in the hydrophobic part of unilamellar liposomal DMPC vesicles. Liposomes (100 µg/mL, PBS, pH 7.4) were labeled with the DPH fluorescent probe at a final concentration of 1 µM, and then the flavonoids were added and incubated at 25 °C for 30 min. Fluorescence intensity values were registered in the absence (F_0_) and in the presence (F) of the flavonoids.

**Figure 8 molecules-28-01252-f008:**
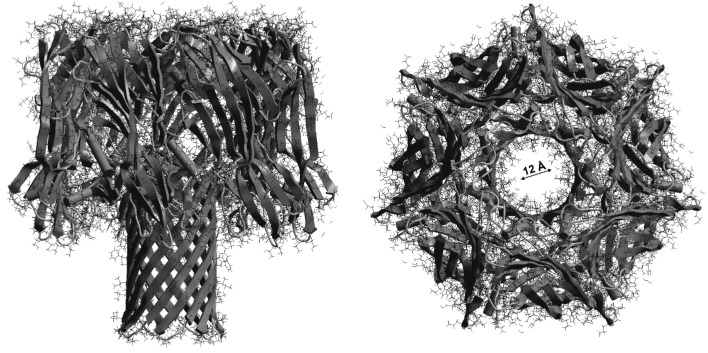
Molecular structure of *S. aureus*
*α*-hemolysin-assembled pore consisting of seven monomers. The diameter of the formed inner cavity is ≈ 12 Å.

**Table 1 molecules-28-01252-t001:** Antimicrobial activities, interaction with liposomal membrane, water solubility (20–25 °C) [41,42,43] and calculated molecular parameters of the flavonoids studied.

Parameter	Naringenin	Quercetin	(+)-Catechin
Minimal inhibitory concentrations (MICs), μM	200	100	150
Dipole moments, D	1.602	0.986	2.107
Torsion angles (C3-C2-C1′-C2′)	86	180	118
Stern–Volmer constants of DPH fluorescence quenching in liposomal membranes, µM^−1^	0.012 ± 0.003	1.66 ± 0.20	0.0012 ± 0.0002
Water solubility, mg/L	4.38	0.51	2260

**Table 2 molecules-28-01252-t002:** Minimal inhibitory concentrations of the flavonoids, MICs, for preventing bacteria cell growth.

Flavonoid	MICs	Bacterial Strain	References
Quercetin	120 µM (35.76 µg/mL) > 3410 µM (> 1024 μg/mL) 830 µM (250 µg/mL) 1670 µM (500 µg/mL) > 3330 µM (> 1000 µg/mL) 1670 µM (500 µg/mL) 170 µM (50 µg/mL) 200 μM (60 µg/mL)	*E. coli* *S. aureus* strain Newman methicillin-susceptible *S. aureus* methicillin-resistant *S. aureus* *S. aureus* *E. coli* *S. aureus* *P. gingivalis*	[52] [20] [53] [54] [55] [55] [56] [57]
Naringenin Glycoside naringin	460 µM (125 μg/mL) 1720 µM (1000 µg/mL)	methicillin-resistant *S. aureus* *S. aureus*	[21] [56]
Catechin	3550 µM (1000 µg/mL)	*S. aureus*	[56]

## Data Availability

Not applicable.

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
