# Peer review of "Antimicrobial Activity of Quercetin, Naringenin and Catechin: Flavonoids Inhibit Staphylococcus aureus-Induced Hemolysis and Modify Membranes of Bacteria and Erythrocytes"

_molecules, 2023, doi:10.3390/molecules28031252_

Round 1

Reviewer 1 Report

I suggest to attend some details in the manuscript:

In line 112, there is a period that may not be there.

In lines 230 and 234 it is suggested to eliminate "the" before "probes"

In line 329 eliminate space between gro and wth

In line 367 replace "propon" with proton

In line 553 replace was added with were added

In line 566, replace 8,1 mM with 8.1 mM

Author Response

Response to Reviewer 1 Comments:

We are very grateful to the Reviewer for the evaluation and critical analysis of our manuscript.

We completely agree with the Reviewer.

We have edited the text according the Reviewer’s suggestions.

In line 112, there is a period that may not be there

We have omitted this period.

In lines 230 and 234 it is suggested to eliminate "the" before "probes"

(lines 231 and 236) We have eliminated “the” before “probes”.

In line 329 eliminate space between gro and wth

(line 331 and 332) We have eliminated space. 

In  line 367 replace "propon" with proton

(line 369) We have replaced "propon" with proton.

In line 553 replace was added with were added

(line 560) We replaced “was added” with “were added”.

In line 566, replace 8,1 mM with 8.1 mM

(line 573) We replace “8,1 mM” with “8.1 mM”.

Reviewer 2 Report

Major concerns:

The novelty of the study is lacked, and the methodology applied are out of date and not enough to support the main conclusion that the agents target either the membrane of sheep cells or mycobacterium.

Other issues:

The figures are bad in presenting and plenties of errors typing, as well as messy format.

Author Response

Response to Reviewer 2 Comments:

We are very grateful to the Reviewer for the evaluation and critical analysis of our manuscript.

The English in this manuscript has been carefully edited by a professional science language editor.

Major concerns:

The novelty of the study is lacked, and the methodology applied are out of date and not enough to support the main conclusion that the agents target either the membrane of sheep cells or mycobacterium.

The novelty of this study consists in comparison of the antihemolytic activity of the flavonoids against hemolysis induced by α-hemolysin and the membrane-modifying action of the flavonoids. The results showed that the antibacterial activities of the flavonoids were determined by disorder in the structural organization of bacterial cell membranes and an increase in the resistance of the target cells (erythrocytes) to α-hemolysin due to modification of the erythrocyte membrane structure. We confirmed that the cell membrane could be one of the direct modes for antibacterial action of the flavonoids. Quercetin, catechin and naringenin exerted different effects on the membrane structure and the size of bacterial cells. The accessibility of naringenin and catechin to the membrane interior was much lower when compared to quercetin in accordance with flavonoid lipophilicity, as revealed by fluorescence studies. We demonstrated that the antihemolytic effect of quercetin correlated with the effect of the flavonoid on the organization and rigidity of the erythrocyte membrane lipid bilayer, which, in turn, inhibited αHL - induced osmotic hemolysis due to prevention of αHL incorporation into erythrocyte membrane.

To gain a deeper insight into the antibacterial activities of the flavonoids or their protective effect on erythrocytes, we carried out a comparative analysis of the effects of the flavonoids on the diameter, Zeta-potential and membrane organization of viable S. aureus cells and sheep erythrocytes. We studied the effect of these compounds on the structural organization of bacterial and erythrocyte membranes, measuring the fluorescence anisotropy values for the DPH and TMA-DPH probes which differ in their membrane localization. The fluorescence anisotropy values for both probes characterize the state of membranes in the inner hydrophobic and in the aqueous/membrane interface regions, respectively.

Using the Stern-Volmer constants, KSV, of DPH fluorescence quenching by the flavonoids in the liposomal membrane, we estimated the availability of the quencher (flavonoid) to the excited fluorophore (DPH) and therefore, flavonoid penetration into the membrane interior.

These methods are widely used to study biochemical and biophysical processes occurring in biological membranes and to evaluate the interactions of the flavonoids with artificial or biological membranes [Kyrychenko A. Using fluorescence for studies of biological membranes: a review. Methods Appl Fluoresc. 2015 Oct 1;3(4):042003. doi: 10.1088/2050-6120/3/4/042003].

Zeta-potential is an electrokinetic potential associated with the mobility of charged particles, and its changes make it possible to estimate electrostatic forces between bacterial cells. The S. aureus cell diameter (nanosize) was analyzed using dynamic light scattering. These methods allowed us to analyze the details of direct interaction of the flavonoids and the cells studied. We demonstrated that quercetin, and not naringenin or catechin, increased the bacterial cell nano-size measured, probably as a result of cell aggregation. Effect of the flavonoids on Zeta potential of bacterial cells, as a possible marker of the changes in the membrane permeability, was not statistically significant.

We edited the Abstract to emphasize the mechanism of antihemolytic effect of quercetin.

We noted that the antihemolytic effect of quercetin was related to the effect of the flavonoid on the organization of the erythrocyte membrane lipid bilayer which, in turn, inhibited α-hemolysin - induced osmotic hemolysis due to prevention of α-hemolysin incorporation into target membrane.

Staphylococcus aureus is a gram-positive cocci in clusters.

Mycobacteria are Gram-positive, non-spore forming rod-shaped bacteria.

The figures are bad in presenting and plenties of errors typing, as well as messy format.

We have checked up the figure presentations and format and redrawn Figure 5.

Reviewer 3 Report

The manuscript describes how three flavonoids (quercetin, naringenin, and catechin) inhibit S. aureus growth (antibacterial activity) and hemolytic activity. The study is of interest, but some points should be revised, as follows:

-      Line 112: Please revise "...combating bacteria. topical..." because this passage is confusing.

-      Line 164: It is necessary to tell the readers what indicates lipophilicity. Based on the text, this information is in Table 1.

-      Regarding the data shown in Figure 2, why the concentrations tested for the three flavonoids were the same if they presented distinct MIC values? If quercetin MIC value was tested, but not the MIC values of the other two flavonoids, would the data presented be biased towards quercetin?

-      Lines 211-212: There were no differences in bacterial Zeta-potencial, but how about bacterial hydrophobicity?

-      Figure 4: Regarding the data shown in Figure 4, why were the concentrations tested for the three flavonoids the same if they presented distinct MIC values? If values close to quercetin MIC value were tested but not close to the MIC values of the other two flavonoids, would the data presented be biased towards quercetin?

-      Table 2: It would be easier for a junior reader if the concentration of flavonoids were presented with the same units (μg/mL or μM). In the Results of the current study, the data are in μM.

-      Line 530: Please state the purity of naringenin, quercetin, catechin. Also, it would be helpful to the reader to have the catalog numbers to access the manufacturer information because there is more than one product available per compound.

-      Lines 537-538: Please state the final concentrations of ethanol used in the assays.

-      Line 554: Please correct "μm".

-      Lines 556-557: Please state whether naringenin, quercetin, and catechin led to turbidity when these compounds were added to the cultures. Many compounds can form complexes with culture medium components, impairing visual observation and turbidity/absorbency reading of microbial growth.

-      Lines 652-653: this text could be removed.

Additional points for review:

Line 78: It is necessary to italicize S. aureus at the beginning of this line.

Line 84: It is necessary to italicize E. coli.

Line 329: It is necessary to correct the word "growth".

Line 457: Please correct the spelling for "perposes". It should be "purposes".

Lines 516, 542, and 550: It is necessary to italicize S. aureus.

Line 586: Please correct "to1.0 μM"by adding a space after "to".

Author Response

Response to Reviewer 3 Comments:

We are very grateful to the Reviewer for the evaluation and critical analysis of our manuscript.

Line 112: Please revise "...combating bacteria. topical..." because this passage is confusing.

Line 112 Should be "...combating bacteria topical..."

Line 164: It is necessary to tell the readers what indicates lipophilicity. Based on the text, this information is in Table 1.

A substance is lipophilic if it is able to dissolve much more easily in lipids than in water. We added to text that lipophilicity refers to the tendency of a compound to partition between a lipophilic organic phase and a polar aqueous phase (Table 1).

Regarding the data shown in Figure 2, why the concentrations tested for the three flavonoids were the same if they presented distinct MIC values? If quercetin MIC value was tested, but not the MIC values of the other two flavonoids, would the data presented be biased towards quercetin?

Effects of quercetin, catechin and naringenin on S. aureus cell diameter and zeta-potential were measured using higher concentrations of flavonoids (up to 200 µM), but catechin and naringenin did not influence these parameters.

Lines 211-212: There were no differences in bacterial Zeta-potential, but how about bacterial hydrophobicity?

We did not measure bacterial cell surface hydrophobicity.

Figure 4: Regarding the data shown in Figure 4, why were the concentrations tested for the three flavonoids the same if they presented distinct MIC values? If values close to quercetin MIC value were tested but not close to the MIC values of the other two flavonoids, would the data presented be biased towards quercetin?

We tested higher concentrations of flavonoids (up to 200 µM) in this experiment, but naringenin and catechin did not inhibit sheep erythrocyte lysis induced by S. aureus.

Table 2: It would be easier for a junior reader if the concentration of flavonoids were presented with the same units (μg/mL or μM). In the Results of the current study, the data are in μM.

We expressed the flavonoid concentrations both as µM and µg/ml in Table 2.

Quercetin

120 µM (35.76 µg/ml)

> 3410 µM (> 1024 μg/mL)

830 µM (250 µg/ml)                                       

1670 µM (500 µg/ml)

> 3330 µM (> 1000 µg/ml)

1670 µM (500 µg/ml)

170 µM (50 µg/ml)

200 μM (60 µg/ml)

Naringenin

460 µM (125 μg/ml)

370 µM (100 µg/ml)

Glycoside Naringin

1720 µM (1000 µg/ml)

Catechin

3550 µM (1000 µg/ml)

Line 530: Please state the purity of naringenin, quercetin, catechin. Also, it would be helpful to the reader to have the catalog numbers to access the manufacturer information because there is more than one product available per compound.

We indicated the purity and the product numbers of naringenin, quercetin, catechin.

Naringenin (98%) Product No. W530098

Quercetin (≥ 95%) Product No. Q4951

(+)-Catechin hydrate (≥ 98%) Product No. C1251

Lines 537-538: Please state the final concentrations of ethanol used in the assays.

The maximal final concentrations of ethanol did not exceed 1-2%.

Line 554: Please correct "μm".

We corrected the unit (μM).

Lines 556-557: Please state whether naringenin, quercetin, and catechin led to turbidity when these compounds were added to the cultures. Many compounds can form complexes with culture medium components, impairing visual observation and turbidity/absorbency reading of microbial growth.

To avoid the effect of flavonoids on turbidity/absorbance of the medium, negative controls were run for each flavonoid concentration. We indicated this in the revised version of MS

Lines 652-653: this text could be removed.

We omitted this text.

Additional points for review:

Line 78: It is necessary to italicize S. aureus at the beginning of this line.

Line 84: It is necessary to italicize E. coli.

Line 329: It is necessary to correct the word "growth".

Line 457: Please correct the spelling for "perposes". It should be "purposes".

Lines 516, 542, and 550: It is necessary to italicize S. aureus.

Line 586: Please correct "to1.0 μM"by adding a space after "to".

We have edited the text according to the Reviewer’s suggestions.